# Characterizing the Water Storage Variation of Kusai Lake by Constructing Time Series from Multisource Remote Sensing Data

Zhengkai Huang [1], Xin Wu [1], Haihong Wang [2,3,*], Zehui Zhao [1], Liting Du [1], Xiaoxing He [4] and Hangyu Zhou [5]

[1] School of Transportation Engineering, East China Jiaotong Univeristy, Nanchang 330013, China; zhkhuang@whu.edu.cn (Z.H.); xwu@ecjtu.edu.cn (X.W.); zhzhao@ecjtu.edu.cn (Z.Z.); ltdu@ecjtu.edu.cn (L.D.)
[2] School of Geodesy and Geomatics, Wuhan University, Wuhan 430079, China
[3] Key Laboratory of Marine Environmental Survey Technology and Application, Ministry of Natural Resources, Guangzhou 510300, China
[4] School of Civil and Surveying & Mapping Engineering, Jiangxi University of Science and Technology, Ganzhou 341000, China; xxh@jxust.edu.cn
[5] School of Computer Science & School of Cyberspace Science, Xiangtan University, Xiangtan 411105, China; zhy@xtu.edu.cn
* Correspondence: hhwang@sgg.whu.edu.cn

**Abstract:** In September 2011, Zhuonai Lake (ZL) in the northeast of Hoh Xil (HX) on the Qinghai–Tibet Plateau (QTP) broke out. The outburst event seriously changed the environmental hydraulics in this region. Due to the insufficient temporal resolution of observations, it is challenging to assess the impact of this event on short-period variations of water volumes in three lakes downstream of ZL. Combining multisource remote sensing data, we constructed long and high-temporal-resolution time series for the lake level, area, and lake water storage (LWS) of Kusai Lake (KL) to characterize the variabilities before and after the outburst. The water level, area, and LWS time series contain 1051 samples from 1990 to 2022, with uncertainties of 0.16 m, 2.5 km$^2$, and 0.016 km$^3$, respectively. The accuracies verified using the Database for Hydrological Time Series of Inland Waters (DAHITI) are 0.26 m, 2.64 km$^2$, and 0.08 km$^3$ for water level, area, and LWS, respectively. We characterized the LWS variations during the past 30 years based on the high temporal resolution LWS time series. Before the outburst, the 1-year and 3.5-year variations dominated the LWS time series, and there was no obvious semi-annual signal. After the outburst, the 3.5-year variation disappeared, and a strong semi-annual oscillation was observed. From 2012 to 2015, the periodic LWS variations in KL were disturbed by the ZL outburst and the subsequent outflow of KL led by the outburst. Regular cyclic signals have been restored since 2016, with an amplified annual fluctuation. By analysis, precipitation, evaporation, and glacier area change are excluded as driving factors of the pattern change in LWS variations of KL. It can be concluded that the altered recharge pattern of KL triggered by the outburst directly resulted in the observed changes in TWS behavior. For the first time, we identified the periodic patterns of LWS variations of KL during the past 30 years and revealed that the ZL outburst event significantly influenced these patterns. This finding contributes to the comprehensive understanding of the effects of the ZL outburst on downstream lake dynamics. Furthermore, the presented procedure for constructing long and high-resolution time series of LWS allows for monitoring and characterizing the short-period variabilities of Tibetan lakes that lack hydrological data.

**Keywords:** Kusai Lake; outburst; satellite altimetry; Landsat images; lake water storage

## 1. Introduction

In the context of warming and increasing humidity in the Qinghai–Tibetan Plateau (QTP), several notable changes have been observed, including heightened precipitation, reduced evaporation, and intensified glacier melting [1–3]. Over the past three decades,

lakes on the QTP have generally experienced a significant expansion [4,5]. In particular, numerous lakes in the northern region of the plateau have witnessed substantial increases in water level, resulting in a high potential risk of geological hazards [6]. In September 2011, a rare lake outburst occurred over Zhuonai Lake (ZL) in the northeastern Hoh Xil (HX) basin. This event significantly altered the hydrological conditions and the ecological environment of the Zhuonai Lake–Salt Lake (ZL–SL) basin, consisting of ZL, Kusai Lake (KL), Haiding Nor (HN), and SL. Overflow floods threaten downstream national infrastructure, such as railroads, highways, and optical cables [7–9]. It is also the first large-scale lake outburst phenomenon observed in the QTP by satellite technology and a typical case of natural disaster triggered by climate change on the QTP. Therefore, the impact of the outburst has received widespread attention from both the public and the scientific community [10–12].

Studies concerning the ZL outburst event primarily relied on remote sensing technologies, such as satellite imagery and satellite altimetry. The former can be used to extract changes in the lake area, while the latter can monitor changes in lake level. Based on satellite images, a sudden decrease in the area of ZL and a subsequent rapid increase in that of KL were observed due to the outburst in September 2011 [13,14]. Correspondingly, a remarkable rise in the water level in KL was also detected using satellite altimetry [15,16]. By integrating the two techniques, lake volume changes in the ZL–SL basin were estimated. For example, Hwang et al. assessed lake volume changes in ZL, KL, and SL using two Landsat-7 images in order to validate results from altimetry data [17]. Aided by the densified water level series, Li et al. computed lake storage changes in the three lakes, demonstrating the possibility for lake overflow flood monitoring [18].

At present, the main challenge in monitoring lake changes on the QTP is the insufficient resolution of satellite observations. Satellite altimetry can provide a minimum revisit period of about 10 days, but the large cross-track distance results in sparse coverage. For lakes on the QTP, even if a satellite track passes through, the rugged terrain around lakes may prevent altimeters from locking echoes and obtaining effective observations [19,20]. In order to retrieve more accurate altimetry data over nearshore and inland waters, numerous efforts have been made, such as data editing [21], optimizing geophysical corrections [22,23], waveform modification, and retracking [24–26]. While these efforts can enhance the quantity and quality of data to some extent, it remains challenging to fully address the limitations of satellite mission configuration. The shortage of remote sensing imagery for extracting lake areas on the QTP can be attributed to several factors. Remote sensing imagery is susceptible to cloud cover. Although many methods have been developed to extract lake boundaries from cloud-obscured images [27–29], the presence of high mountains and extensive cloud cover in plateau regions hinder the acquisition of clear and uninterrupted imagery, causing limited coverage and availability of suitable images for lake boundary extraction [30,31]. The temporal resolution of the remote sensing data may also pose a limitation. Regular and frequent satellite acquisitions are necessary to monitor dynamic changes in lake boundaries. However, the availability of consistent and continuous temporal sampling can be limited, making it difficult to capture detailed features of lake area variations, such as seasonal and annual fluctuations. Therefore, most current studies focus on detecting abrupt changes corresponding to the outburst and long-term trends in the ZL–SL basin [32].

The fusion of multisource data is an effective way to improve temporal resolution. Li et al. constructed high temporal resolution water level and storage change data sets for 52 large lakes on the Tibetan Plateau from 2000 to 2017 using multiple altimetric missions and Landsat-derived lake shoreline positions [33]. By establishing the relationship between water level and area, they generated a time series of lake water storage (LWS) with temporal resolution ranging from one week to one month. Combined with Cryosat-2 data and Landsat images, Wang et al. determined the monthly time series of LWS from 2009 to 2016 for four lakes in the ZL–SL basin, revealing the impact of the outburst on the hydrological system [34]. These studies showcase the potential for monitoring short-period dynamic changes of lakes in the QTP.

This study aims to explore the influence of the 2011 outburst event on the local hydrological environment in the ZL–SL basin. A procedure for constructing a time series of LWS is presented by combining satellite altimetry data, satellite images, and a digital elevation model (DEM). We focused on KL, which has much better conditions than the other three lakes in the ZL–SL basin, to investigate the LWS time series before and after the outburst. Several altimetry missions, such as the Jason family, ERS (European Remote Sensing Satellite) family, ICESat (Ice, Cloud, and land Elevation Satellite), and Cryosat-2, have been tracked over KL, providing lake-level observations since the 1990s. Landsat satellites have captured two scenes covering KL (Path 137/Row 035 and Path 138/Row 035). The abundance of satellite observations presents an opportunity to comprehensively characterize the variations in KL's water storage, thus enhancing our understanding of the impact of the 2011 outburst event. The paper is structured as follows: Section 2 provides an overview of the study area and the data sources. Section 3 explains the procedures and methods used for data processing and the construction of high-resolution LWS time series. In Section 4, the results are presented and validated. Section 5 discusses the characteristics of LWS changes (ΔLWS) in KL and examines the influence of the outburst event. Finally, the conclusions are summarized in Section 6.

## 2. Materials

### 2.1. Study Area

KL (35.5°N–35.8°N, 92.5°E–93.25°E), located in the northeastern part of the Hoh Xil National Nature Reserve in the central QTP, is a typical plateau closed lake predominantly fed by precipitation and snowmelt, with evaporation serving as the primary mechanism for water loss. It is the largest subbasin in the ZL–SL basin, with a catchment area of 4132 km$^2$, as depicted in Figure 1. Within the KL basin, an estimated area of 10–15 km$^2$ is covered by mountain glaciers. The region has a cold and arid climate, with an annual average precipitation of 294.2 mm and an annual average temperature of −4.2 °C [35]. The water level and area of KL obtained using satellite technology in 2000 were ~4477 m and ~260 km$^2$, respectively, which experience seasonal fluctuations due to the distinct wet and dry seasons. The lake plays a critical role in maintaining the fragile ecosystem of the Reserve. The outburst of ZL in 2011 has fundamentally altered the water balance of KL, causing it to receive water from ZL and release excess water into downstream lakes [36]. Since 2012, the lake has covered more than 320 km$^2$. Given its remote location and ecological significance, KL offers an intriguing research area to study the hydrological dynamics, water storage variations, and the impact of climate changes on this pristine alpine lake ecosystem.

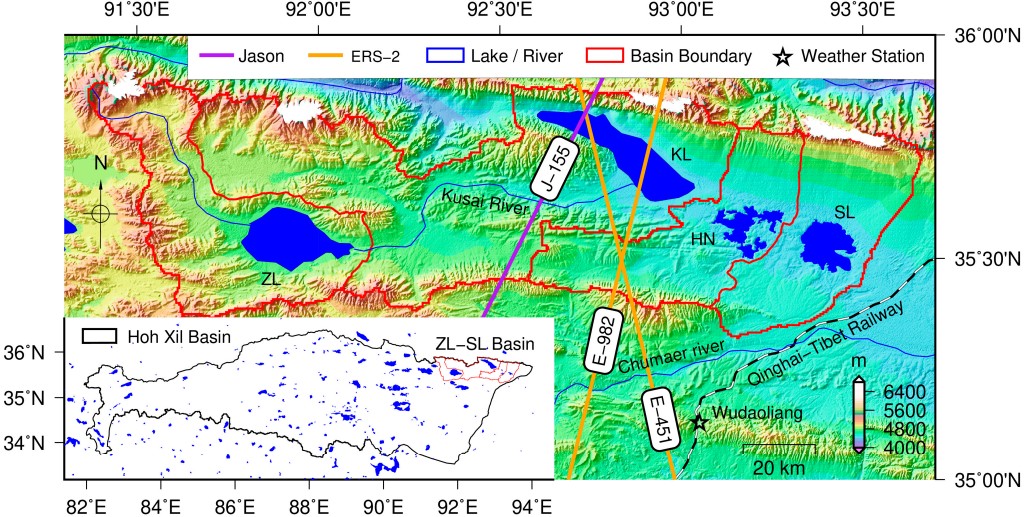

**Figure 1.** The ZL–SL basin location and satellite ground tracks over Kusai Lake. KL: Kusai Lake, ZL: Zhuonai Lake, HN: Haiding Nor, SL: Salt Lake.

## 2.2. Data

### 2.2.1. Satellite Altimeter Data

The altimeter data used in this study were obtained from the Jason family satellites and ERS-2 (European Remote Sensing Satellite-2). Table 1 provides selected information about these satellites. The Jason family satellites, including Jason-1, Jason-2, and Jason-3, have been providing continuous observation with a repeat cycle of 9.91 days from February 2002 to the present. The ERS-2 satellite was an altimetry mission launched by the European Space Agency (ESA) in April 1995. ERS-2 operated on a 35-day orbit cycle, enabling it to cover a wide range of global locations. It provided valuable data for monitoring lake-level changes on the QTP. As illustrated in Figure 1, Jason-1/2/3 has one track, and ERS-2 has two tracks flying over KL. This study utilizes Sensor Geophysical Data Record (SGDR) products, which enable customized waveform retracking to retrieve more accurate lake levels. Jason-1/2/3 data were downloaded from the French Space Agency AVISO (ftp://ftp-access.aviso.altimetry.fr, accessed on 13 May 2022), and ERS-2 data were downloaded from the ESA ftp site (ftp://ra-ftp-ds.eo.esa.int, accessed on 28 August 2016).

**Table 1.** Information on satellite altimeter data used in this study.

| Satellite | Altimeter | Along-Track Resolution (20 Hz) | Repeat Cycle | Time Span | Available Cycles |
|---|---|---|---|---|---|
| ERS-2 | RA-1 | 330 m | 35 d | May 1995–February 2003 | 78 |
| Jason-1 | Poseidon-2 | 290 m | 9.91 d | February 2002–April 2011 | 100 |
| Jason-2 | Poseidon-3 | 290 m | 9.91 d | October 2008–May 2017 | 186 |
| Jason-3 | Poseidon-3B | 290 m | 9.91 d | March 2016–April 2022 | 208 |

### 2.2.2. Remote Sensing Images

Remote sensing images with a spatial resolution of 30 m from Landsat-5/7/8 were used to extract the lake area of KL. These images were obtained from the United States Geological Survey (USGS) via https://earthexplorer.usgs.gov (accessed on 31 August 2022). Images with cloud cover exceeding 20% were excluded, and those with water body boundaries that were difficult to interpret accurately were also removed. Ultimately, we acquired 684 usable Landsat images from 1990 to 2022, as outlined in Table 2.

**Table 2.** Overview of Landsat family remote sensing images utilized in this paper.

| Satellite | Spatial Resolution | Repeat Cycle | Band | Time Span | No. of Images |
|---|---|---|---|---|---|
| Landsat-5 | 30 m | 16 d | Green\NIR(Band2\Band4) | January 1990–October 2011 | 275 |
| Landsat-7 | 30 m | 16 d | Green\NIR(Band2\Band4) | September 1999–August 2021 | 275 |
| Landsat-8 | 30 m | 16 d | Green\NIR(Band3\Band5) | May 2013–November 2021 | 134 |

### 2.2.3. Topographic Data

Comprehensive topographic data are required to establish the lake level and area relationship. We integrated surface topography and underwater topography to ensure the relationship encompasses the entire KL basin. Surface topography data were derived from the Shuttle Radar Topography Mission (SRTM) with a $1''$ resolution, sourced from the United States Geological Survey (USGS) via https://earthexplorer.usgs.gov/ (accessed on 31 August 2022). The accuracy of SRTM in the QTP is far better than the nominal

accuracy, up to 3.33 m in some flat regions [37]. KL had a lake level of 4475 m and an area of 259.48 km$^2$ as determined by SRTM data, which were originally collected from 11 February 2000 to 22 February 2000 [38]. This result is highly consistent with the lake level and area derived from satellite altimetry and remote sensing images during the same period [34]. It is indicated that the water level and area data obtained from SRTM are reliable [39]. The underwater topography is obtained by digitizing the bathymetric map with a contour interval of 5 m, which is published in reference [40].

### 2.2.4. Ancillary Data

The precipitation and evaporation data from Wudaoliang (93.05°E, 35.13°N), the nearest meteorological station available, were utilized to discuss the LWS variations. Daily datasets covering the period from 1990 to 2017 were sourced from the China Meteorological Data Sharing Network at http://data.cma.cn (accessed on 31 December 2020). Additionally, glacier area data within the KL basin were obtained using the Normalized Difference Snow Index (NDSI) applied to Landsat images.

## 3. Methods

The technical workflow of this study is illustrated in Figure 2. Firstly, we derived lake areas and levels of KL from Landsat images and satellite altimeter data. A discrete lake level-area (LA) curve was constructed based on topographic data and calibrated using reliable satellite observations. Secondly, armed with the corrected LA curve, we conducted mutual interpolation between satellite-derived lake levels and areas to densify each other. Finally, a high temporal resolution LWS time series for KL was determined using densified level and area datasets. Results were validated by crosschecking with independent water level datasets published internationally.

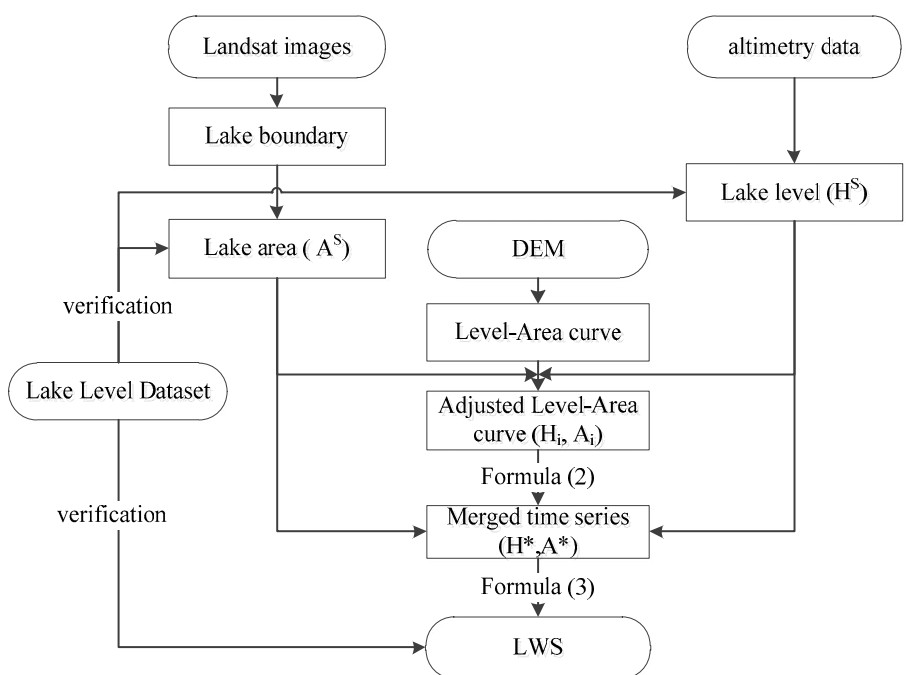

**Figure 2.** The flowchart for the construction of high temporal resolution lake water storage time series. *A\** and *H\** denote the interpolated lake area and water level values.

### 3.1. Monitoring Lake Changes Based on Satellite Technology

#### 3.1.1. Estimation of Lake Areas from Landsat Images

In this study, the Normalized Difference Water Index (NDWI) was selected as the method to extract the lake water boundaries from the Landsat images. The NDWI is widely used to normalize specific bands of the Landsat images and is applied to distinguish

between land and water bodies with appropriate thresholds [41]. The calculation formula for NDWI is as follows:

$$NDWI = (NIR - G)/(NIR + G) \qquad (1)$$

where NIR represents the reflectance of the near-infrared band (band 4 for TM/ETM+, band 5 for OLI/TIRS), and G represents the reflectance of the green band (band 2 for TM/ETM+, band 3 for OLI/TIRS).

For the estimation of the lake area, we used the ArcGIS10.8 software to process the Landsat images. Firstly, images with cloud coverage exceeding 20% were filtered out. Secondly, NDWIs were calculated based on the reflectance values of the green and near-infrared bands using the "Raster Calculator" tool in ArcGIS. Thirdly, the threshold method extracted the water body and its boundary [42]. Pixels with NDWI greater than 0 were marked as water bodies, and pixels with NDWI less than 0 were marked as land. To minimize possible misclassification errors, the extracted lake boundary was manually edited by visual comparison with the original image. Finally, based on the delineated lake boundary, the lake area was obtained using the "Calculate Geometry" tool in the ArcGIS software.

The uncertainty of the estimated lake area can be evaluated using a buffer method with a buffer size of half a pixel [43]. That is, the accuracy of the lake area was obtained by multiplying the total number of margin pixels by half of the pixel area.

### 3.1.2. Extraction of Lake Levels from Altimeter Data

Considering the variation of lake extent, we extracted altimeter data over the lake for each cycle using the lake boundary derived from the Landsat images closest to the observation time of altimetry. A 20% threshold retracker was applied to waveforms. This retracking method is suitable for retrieving more accurate altimeter measurements over seasonally ice-covered lakes [20,44]. Various geophysical corrections are required to derive water level, including atmospheric path delay, solid earth tide, and geoidal gradient correction [22]. To keep tropospheric delay corrections consistent across satellite missions, these corrections should be recalculated using the same climate model [20,45]. This study recalculated the dry and wet troposphere corrections using the most recent climate dataset, ECMWF (European Centre for Medium-Range Weather Forecasts) Reanalysis v5 (ERA5). ERA5 is the fifth generation ECMWF atmospheric reanalysis, produced by the Copernicus Climate Change Service (C3S) at ECMWF [46]. Geoidal gradient corrections were computed using the EGM2008 model. The geoid heights range from −43.98 to −43.42 m. Hence, the effect of the geoidal gradient can reach several decimeters. Other corrections were obtained from SGDR. After correcting for geophysical effects, a robust two-step procedure is employed to estimate the mean lake levels for each cycle. In the first step, the moving Median Absolute Deviation (MAD) filtering is applied to all observations for all cycles, which can effectively remove large outliers in terms of long-time trend change. In the second step, a two-sigma (standard deviation) criterion is recurrently performed to delete outliers cycle by cycle. For detailed information about the two-step procedure, please refer to the publication cited as [20]. The difference between reference ellipsoids of the Jason family and ERS-2 is considered. Deviations between satellite missions also need to be corrected. These deviations were determined using observations in the overlapping period between each two consecutive satellite missions [20].

### 3.2. Construction of Lake Water Level-Area (LA) Curve

The LA curve is important to bridge lake levels by altimetry and areas by satellite imagery. We used SRTM data to extract contour lines within the lake basin, with a contour interval of 1 m and elevations higher than the lake surface. The area values corresponding to these contours were computed to create data pairs, representing the relationship between area and lake level. That is the LA curve above the lake surface derived from SRTM. In order to eliminate the inconsistency of the elevation datum and possible systematic deviation between SRTM and satellite altimetry observations, the SRTM-derived LA curve

should be calibrated using concurrent observations from altimetry and Landsat imagery. The altimeter-derived level and Landsat-derived area observed in a day were paired for the calibration. After calibration, the LA curve for the whole lake basin was obtained by supplementing the bathymetric data.

The LA curve for KL is shown in Figure 3. The black line in Figure 3a represents the SRTM-derived LA curve. The red points represent 51 data pairs obtained from the Jason-2 satellite and Landsat imagery. The bias in elevation is 0.707 m, which can be attributed to the difference between the ellipsoids used for SRTM and Jason satellites, respectively [47]. The blue line in Figure 3a illustrates the calibrated LA curve. The green line in Figure 3b is the complete LA curve of KL obtained by combining the SRTM and bathymetric data.

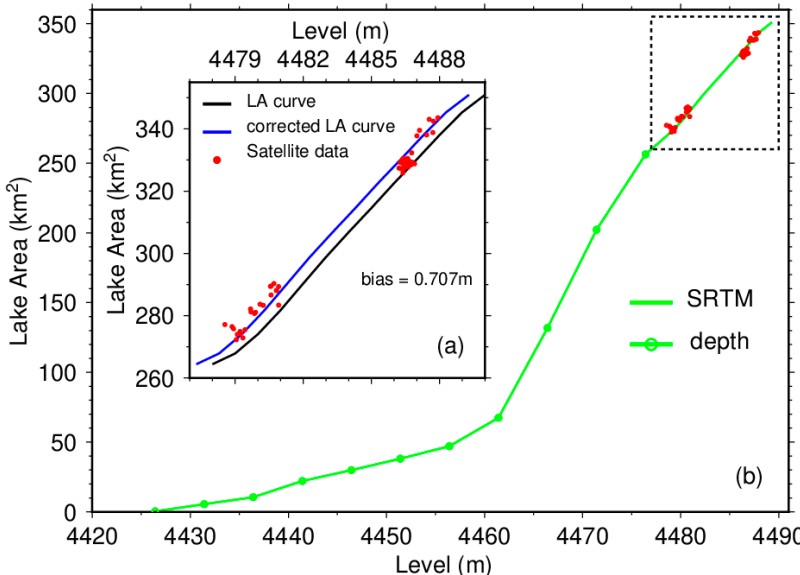

**Figure 3.** The LA curve of Kusai Lake. (**a**) SRTM-derived LA curves above the lake surface before and after calibration. (**b**) The complete LA curve combining SRTM and bathymetry data.

*3.3. Interpolation between Lake Level and Area*

Since the operating times of altimetry satellites and Landsat satellites may not align perfectly, we conducted mutual interpolation of water level and area using the LA curve. This interpolation helped compensate for data gaps arising from the different operational periods of these satellite technologies. The calculation formula is as follows:

$$\begin{cases} A^* = A_{i-1} + (A_i - A_{i-1})\frac{H^S - H_{i-1}}{H_i - H_{i-1}} \\ H^* = H_{i-1} + (H_i - H_{i-1})\frac{A^S - A_{i-1}}{A_i - A_{i-1}} \end{cases} \tag{2}$$

In Equation (2), $H^S$ and $A^S$ are the lake water level and area obtained from satellite technology, while $A_i$ and $H_i$ represent the LA curve's water level and area values. The subscript $i$ is the number of the first value larger than $H^S$ or $A^S$. $A^*$ and $H^*$ denote the interpolated area and water level values corresponding to $H^S$ and $A^S$ obtained by linear interpolation through the LA curve.

*3.4. Computation of Lake Water Storage (LWS)*

The relative LWS is calculated using the following formula [48].

$$\Delta\text{LWS} = \frac{1}{3}(H_1 - H_0) \times \left(A_1 + A_0 + \sqrt{A_1 A_0}\right) \tag{3}$$

where $H_0$ and $H_1$ are the water levels at epochs $t_0$ and $t_1$, $A_0$ and $A_1$ are the corresponding areas, and $\Delta$LWS is the LWS change between the two epochs.

The uncertainty of the estimated ΔLWS can be assessed using the law of error propagation as follows [49].

$$\sigma_{\Delta LWS} = \frac{1}{6}\sqrt{\sigma_{H_0}^2 \left(\frac{\partial \Delta LWS}{\partial H_0}\right)^2 + \sigma_{H_1}^2 \left(\frac{\partial \Delta LWS}{\partial H_1}\right)^2 + \sigma_{A_0}^2 \left(\frac{\partial \Delta LWS}{\partial A_0}\right)^2 + \sigma_{A_1}^2 \left(\frac{\partial \Delta LWS}{\partial A_1}\right)^2} \qquad (4)$$

where $\sigma_{H_i}$ is the uncertainty of altimetric water level, which can be evaluated using the standard deviation of altimeter data in each cycle and $\sigma_{A_i}$ is the uncertainty of the Landsat-derived area.

## 4. Results and Validations

Using the procedure and methods described in Section 3, we built a dataset containing the lake area, water level, and LWS of KL spanning 32 years, based on altimeter data (Jason-1/2/3 and ERS-2), Landsat-5/7/8 images, and topographic data. The results are presented and validated in this section. As no available in situ measurements exist, three previously published datasets are selected as independent validation data. The first dataset, derived from Liao et al., comprises lake levels obtained using multi-altimeter data from 2002 to 2016 [50]. The second is the water level time series retrieved from the Database for Hydrological Time Series of Inland Water (DAHITI), developed by the Deutsches Geodätisches Forschungsinstitut der Technischen Universität München (DGFI-TUM). The water level time series in DAHITI was also derived from multi-mission satellite altimetry [51]. Lastly, Li et al. [33] developed the third dataset, which provides densified water levels and LWS by combining altimetry and optical remote sensing imagery. Table 3 gives a summary of these datasets. Water levels from these datasets were converted into area values using Equation (2) based on the LA curve to validate the lake area. LWS can be calculated directly from water levels using Equations (2) and (3). The specific details are described as follows.

**Table 3.** Datasets used for the validation of results in this study.

| Literature | Dataset | Data Source | Time Range | Density of Data |
|---|---|---|---|---|
| This study | Level/Area/LWS | ERS-2, Jason-1/2/3, Landsat-5/7/8 | January 1990–December 2021 | 35.7/y |
| Liao et al. (2018) [50] | Level/-/- | Envisat, Jason-2, Cryosat-2 | August 2002–January 2017 | 10.6/y |
| DAHITI [51] | Level/-/- | Jason-2/3, Saral, Sentinel-3A | October 2008–August 2022 | 27.6/y |
| Li et al. (2019) [33] | Level/-/LWS | Envisat, Jason-1/2/3, ICESat, Cryosat-2 | February 2000–June 2018 | 38.8/y |

### 4.1. Lake Area

A total of 668 images from 1990 to 2021 from Landsat-5/7/8 were interpreted using the NDWI to derive the lake area of KL. Out of these, 16 images were excluded due to striping errors and cloud cover that prevented the extraction of complete lake boundaries. As a result, we obtained 652 valid area values, as depicted in Figure 4a. To evaluate the accuracy of our results, three methods were employed.

Firstly, buffer analysis with half a pixel buffer size was conducted to estimate the uncertainty [43]. The uncertainties of the lake areas were about 1.7 km² for Landsat-5, 1.9 km² for Landsat-7, and 2.5 km² for Landsat-8, respectively. It indicates that the relative errors of the lake areas derived from Landsat images are all less than 0.8%.

Secondly, we compared the lake areas interpreted from images during the overlapping periods of two adjacent satellites, as shown in Figure 4b,c. The correlation between lake areas interpreted from Landsat-5 and Landsat-7 was 0.99, with a standard deviation (STD) of 1.17 km². Similarly, for Landsat-7 and Landsat-8, the correlation was 0.88, with a corresponding STD of 1.11 km². These findings affirm the relative stability and high consistency of results obtained from Landsat missions.

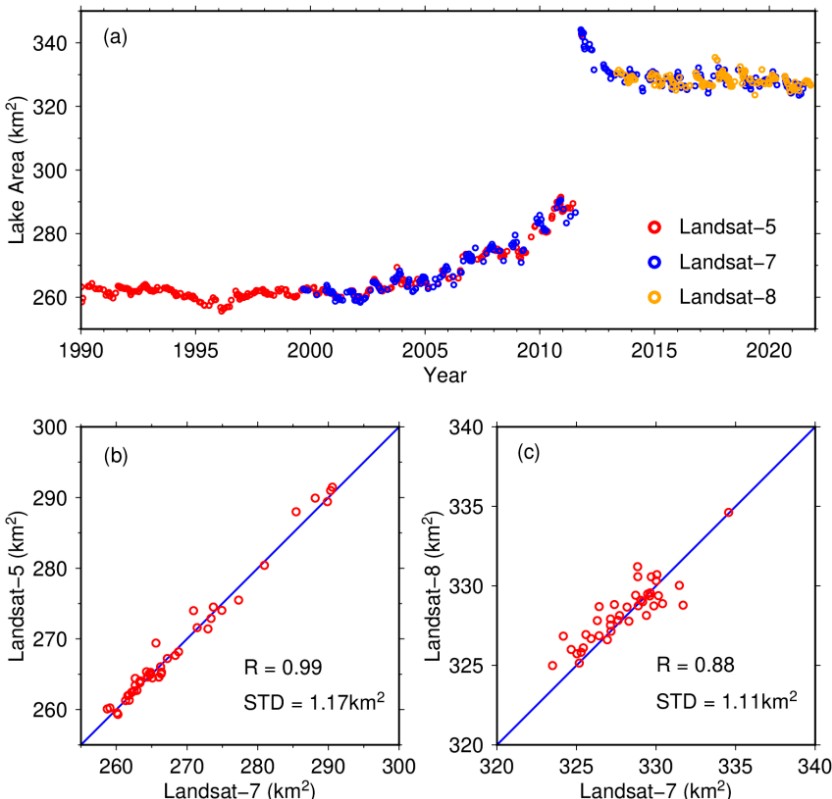

**Figure 4.** The lake area of KL obtained from Landsat images. (**a**) Lake area time series of KL from 1990 to 2021. (**b**) Comparison of the lake area from Landsat-5 and Landsat-7 images. (**c**) Comparison of the lake area from Landsat-7 and Landsat-8 images.

Thirdly, we compared our results with independent validation datasets given in Table 3. The comparison is demonstrated in Figure 5. It can be seen that our results agree best with those from DAHITI, with an STD of 2.64 km$^2$. Compared with Li et al. [33] and Liao et al. [50], the STDs were 4.64 km$^2$ and 4.98 km$^2$, respectively. The relative errors between these datasets were less than 1.4%, and the correlations were close to 1.

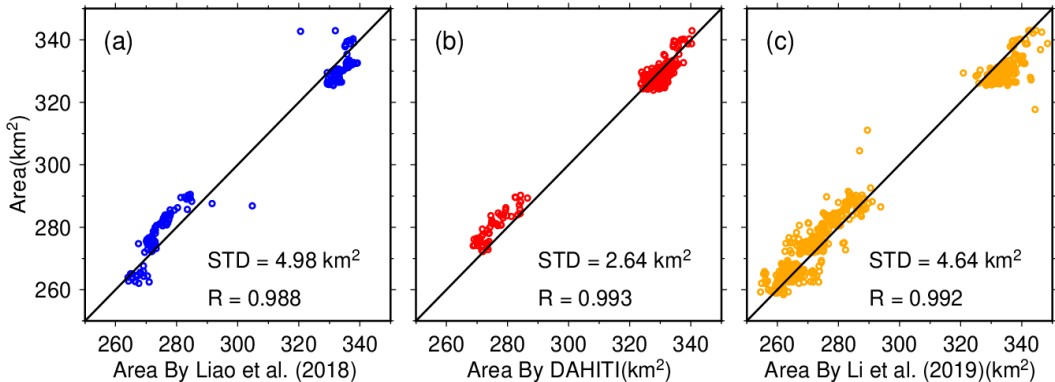

**Figure 5.** Comparison of the lake area time series of KL obtained in this study with (**a**) Liao et al. [50], (**b**) DAHITI [51], and (**c**) Li et al. [33].

The above evaluation results imply that this study's lake area time series is reliable. The STDs obtained from the third method are slightly larger than the first two. This discrepancy can be attributed to errors in the validation datasets used and potential conversion errors resulting from the LA curve.

### 4.2. Lake Level

Lake levels of KL, retrieved from satellite data, are presented in Figure 6. Using multi-altimeter data, a total of 476 lake levels were obtained from May 1995 to April 2022. Figure 6a shows that the altimeter-derived lake level time series is highly correlated with the Landsat-derived lake area time series. The uncertainties of lake levels are 35 cm for ERS-2 and 14 cm, 13 cm, and 7 cm for Jason-1/2/3, respectively, in terms of the root mean square error (RMSE) [52]. Adjustments were made to the biases of ERS-2, Jason-1, and Jason-3, using Jason-2 data as the reference, resulting in biases of 2.60 m, 0.05 m, and 0.55 m, respectively.

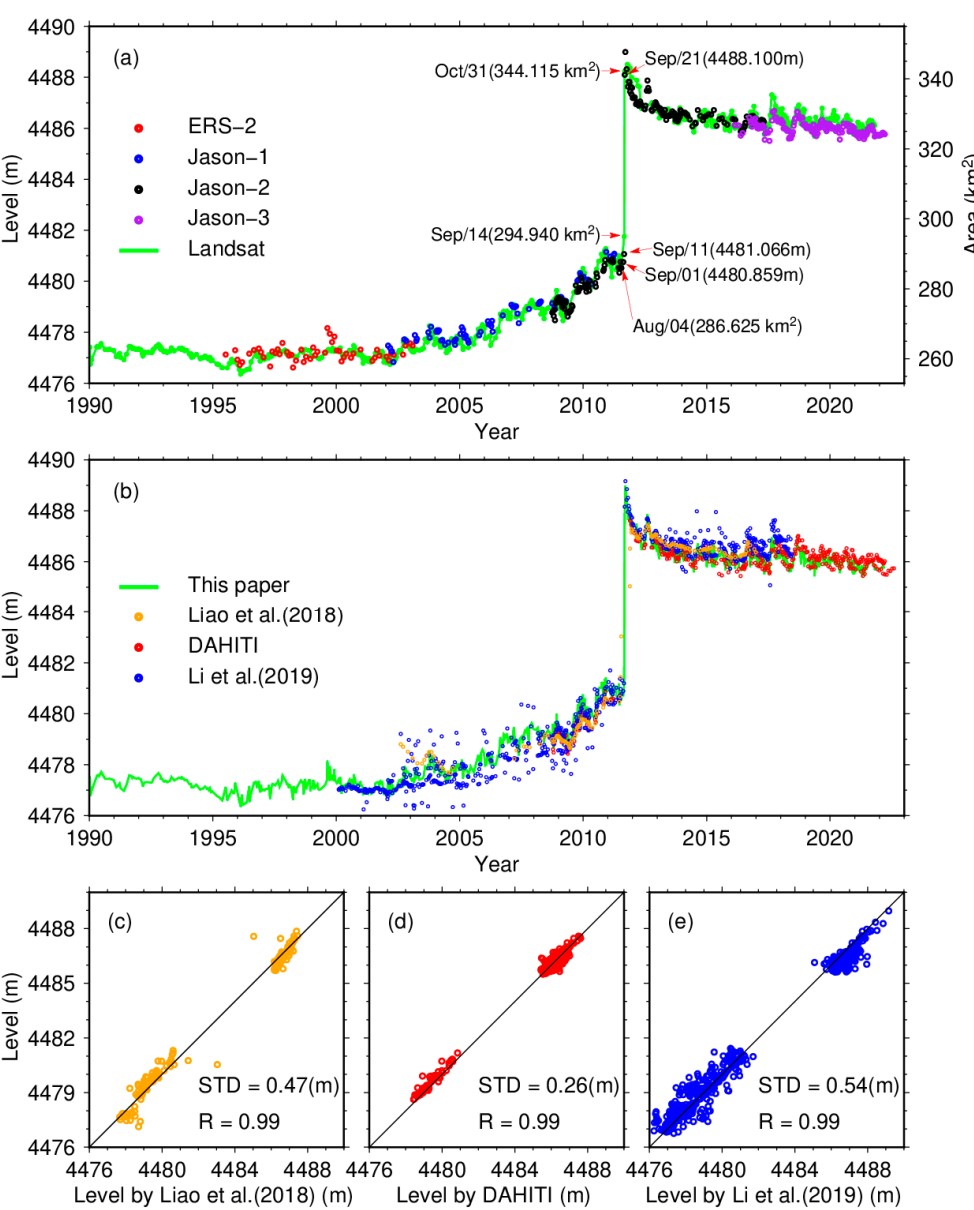

**Figure 6.** Comparison of the lake level time series of KL. (**a**) Altimeter-derived levels and Landsat-derived areas, (**b**) fused lake level time series from this study (green line) and three datasets (colored dots) for validation, and (**c**–**e**) comparison with Liao et al. [50], DAHITI [51] and Li et al. [33], respectively.

Based on the LA curve, the Landsat-derived areas and the altimeter-derived levels were fused to construct an extended and densified lake level time series with 1051 samples from January 1990 to April 2022. The fused lake level time series is illustrated as the green line in Figure 6b, accompanied by three validation datasets in Table 3. Figure 6c–e shows

the comparison between our result and the validation datasets. Similar to the area, our result is the most consistent with DAHITI (red dots), with an STD of 26 cm. The STDs with respect to Liao et al. [50] and Li et al. [33] are much larger. As shown in Figure 6b, the lake levels from Liao et al. (orange dots) and Li et al. (blue dots) are apparently noisy compared to DAHITI. However, the correlation coefficients between our result and the three datasets are all 0.99.

The integration of altimetry data and imagery data efficiently improved the temporal resolution of observations by making up for data gaps that might exist when using a single technique alone. The improvement of temporal resolution, which will be further discussed in the next section, provided a more accurate and comprehensive understanding of the short-term phenomenon and abrupt changes. A good example is the determination of the date of the ZL outburst. In September 2011, the satellite altimetry observed three consecutive water levels over KL: 4480.86 m on 1 September, 4481.01 m on 11 September, and 4488.1 m on 21 September. It indicates that the water lake of KL did not show significant changes before 11 September, whilst there was a rapid and dramatic increase in water level between 11 and 12 September. This rapid and dramatic rise during this specific time period suggests a significant event or factor influencing the water level of KL during that time. Satellite imagery revealed that the area of KL was 286.64 km$^2$ on 5 August, 294.94 km$^2$ on 14 September, and 344.12 km$^2$ on 31 October, respectively, which implies an abnormal expansion in the area occurred before 14 September. Combining results from both satellite observations, it can be inferred that KL received overflow water from ZL between 11 and 14 September 2011. This conclusion drastically narrows down the estimated time range for the outburst of ZL [6,7].

Furthermore, it is worth noting that there was a lack of remote sensing data over Lake KL for 153 days from 26 May to 26 October 2012. During this period, multiple altimetry satellites filled the data gap of satellite imagery with nine observations and effectively detected a bump in water level.

### 4.3. The LWS Time Series of the KL

Combining the mutual interpolations of altimeter-derived water levels and Landsat-derived lake areas, the ΔLWS were computed using Equation (3). The LWS time series of KL was constructed by cumulating ΔLWS, which contained 1051 samples from 22 January 1990 to 3 April 2022. Hence, the LWS is relative to the 22 January 1990 LWS. The average uncertainty of these LWS values is 0.016 km$^3$, estimated by Equation (4).

To evaluate the improvement in resolution after the fusion of two types of satellite observations, we counted the number and interval of samples for each time series from 24 July 1995 to 3 November 2021, when both satellite altimetry and satellite imagery had data coverage. The results are presented in Table 4. Although there was still a 65-day data gap in the LWS time series after the fusion, the average interval had decreased to about 9 days from 16.49 days for the Landsat-derived area time series and 20.47 days for the altimeter-derived level time series. During this period, the data density of the LWS time series is 40/y, which is higher than those of previously published datasets in Table 3.

**Table 4.** Sampling Intervals of the Landsat-Derived Area, Altimeter-Derived Level, and LWS Time Series of KL during 24 July 1995–3 November 2021.

|  | No. of Samples | Average Interval | Min Interval | Max Interval |
|---|---|---|---|---|
| Landsat-derived area | 582 | 16.49 d | 1 d | 153 d (26 May 2012–26 October 2012) |
| Altimeter-derived level | 469 | 20.47 d | 2.38 d | 175 d (8 January 2001–2 July 2001) |
| LWS | 1051 | 9.13 d | 1 d | 65 d (11 August 2008–15 October 2008) |

Figure 7a gives several time series of the LWS of KL from different sources. The LWS time series constructed in this study (hereafter referred to as LWS1) is represented as the black line with blue dots, spanning from 1990 to 2022. The green dots denote the

LWS calculated using water levels from DAHITI, and the purple dots represent the LWS obtained by Li et al. [33]. The comparisons between them are illustrated in Figure 7b,c. The correlation is 0.99 in both cases, indicating good consistency between these results. However, it is evident that the LWS obtained in this study and from DAHITI show a closer agreement compared to the LWS provided by Li et al., which exhibits larger oscillations. The STD between LWS1 and DAHITI is 0.08 km$^3$, and 0.26 km$^3$ between LWS1 and Li et al.

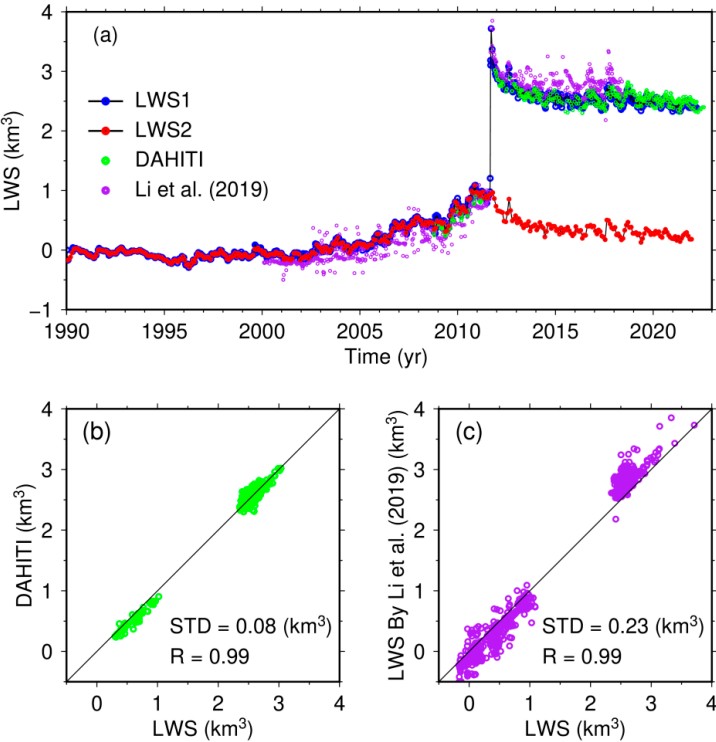

**Figure 7.** Comparison of the LWS of KL from different sources. (**a**) LWS time series obtained in this study (line with dots), from DAHITI (green dots) and Li et al. (purple dots). (**b**) Comparison between LWS1 and DAHITI. (**c**) Comparison between LWS1 and Li et al. [33].

LWS1 reveals three distinctive stages in water volume change for KL. Before 2000, the lake water volume remained relatively stable LWS. A pronounced increase was observed after 2001 until the outburst occurred in ZL in September 2011. Then, the LWS started to decrease in November 2011. It can be found that the LWS rapidly increased by 2.76 km$^3$ in September 2011 and decreased by 0.53 km$^3$ in October 2011. In a normal year, the average variation in September and October was about 0.01 km$^3$ and 0.02 km$^3$, respectively, estimated from 1990 to 2010. These two abnormal water volume changes corresponded to the outburst of ZL and the overflow of KL, respectively [7,9]. To facilitate further analysis, we developed a new LWS time series by removing the abrupt rise due to the outburst in September 2011, which is denoted as LWS2, and demonstrated using the line with red dots in Figure 7a.

The impact of the outburst on the trend of the LWS time series of KL is obvious. Before the outburst, the LWS change trend was observed to be 0.038 km$^3$/y, consistent with the trends observed in other lakes in the HX region [53]. However, after the outburst, the trend shifted to −0.04 km$^3$/y, which is notably different from the surrounding lakes [6].

## 5. Discussion

*5.1. Periodic Variations of the LWS of KL before and after the Outburst*

As mentioned previously, the outburst event in ZL significantly impacted the trend of KL's LWS. To analyze the potential changes in periodic LWS variations caused by the outburst, wavelet and Fourier analyses were conducted on the LWS2 dataset, in which

the abrupt signal was eliminated. These analytical techniques allow for the identification and characterization of periodic variations in time series [19,20,54]. The results of these analyses can provide insights into any changes or disruptions in the periodic patterns of LWS variations following the outburst event.

The wavelet analysis was performed using the wavelet toolbox provided in MATLAB software (Version R2017a). The continuous wavelet transform 'CWT' was applied to the LWS time series. In this study, we chose the Morlet as the analytic wavelet, widely used in geophysical data analysis [19]. The CWT coefficients and wavelet spectra are presented in Figures 8 and 9 for the LWS time series before and after the outburst, respectively. Before the outburst, there were three obvious periodic signals in the LWS time series. Among these signals, the dominant one was the annual signal, followed by an approximately 3.5-year signal. The amplitude of the six-year period signal was the weakest. The amplitude of the 3.5-year cycle signal increased continuously from 1990 to 2011. However, the situation changed dramatically after the outburst, as shown in Figure 9. The 3.5-year signal became too weak to detect, and the annual signal was very vague from 2012 to 2015, which can be attributed to the outflow of KL. It was not until 2016 that the annual signal gradually recovered and reached its strongest in 2019. Notably, a strong semi-annual variation appeared in the LWS after 2012. The comparison between Figures 8 and 9 suggests that the outburst of ZL significantly impacted the periodic variation of KL's water volume. It can be explained by the change in KL's recharge mode. Originally, KL was a typical closed and frozen lake on the QTP. Previous studies have shown that the closed and frozen lakes on the QTP have a weak semi-annual variation [19,20]. However, the outburst connected KL with ZL upstream and HN downstream. The input from ZL and the outflow into HN regulated the water volume in KL after the outburst, which strengthened the semi-annual variation.

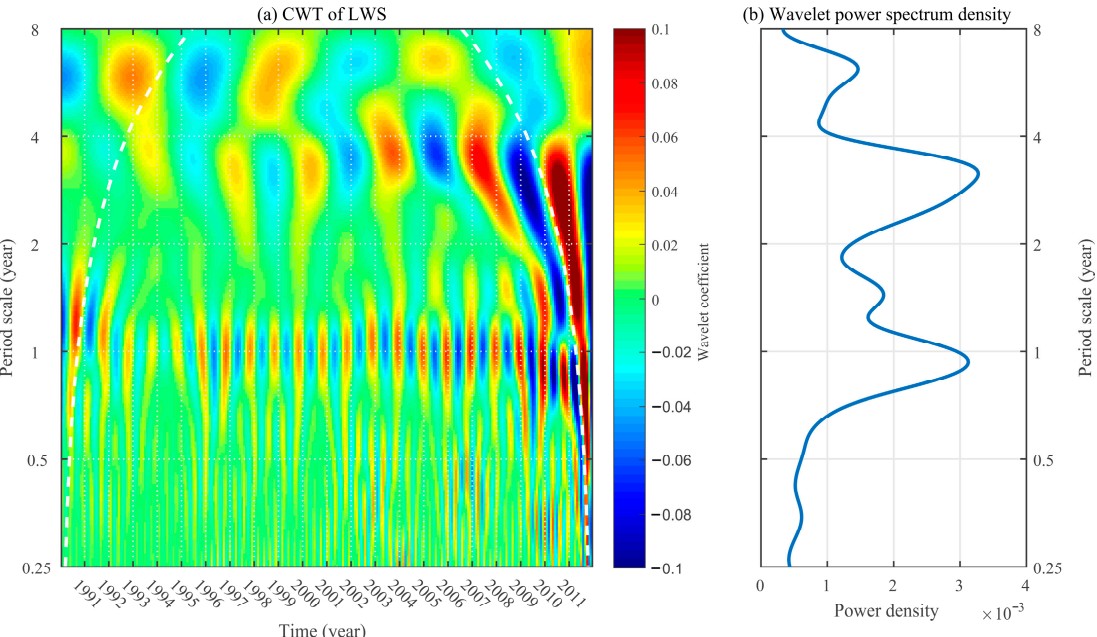

**Figure 8.** Wavelet analysis of the LWS time series of the KL from 1990 to 2011 using the Morlet wavelet: (**a**) continuous wavelet transform coefficient, (**b**) wavelet power spectrum.

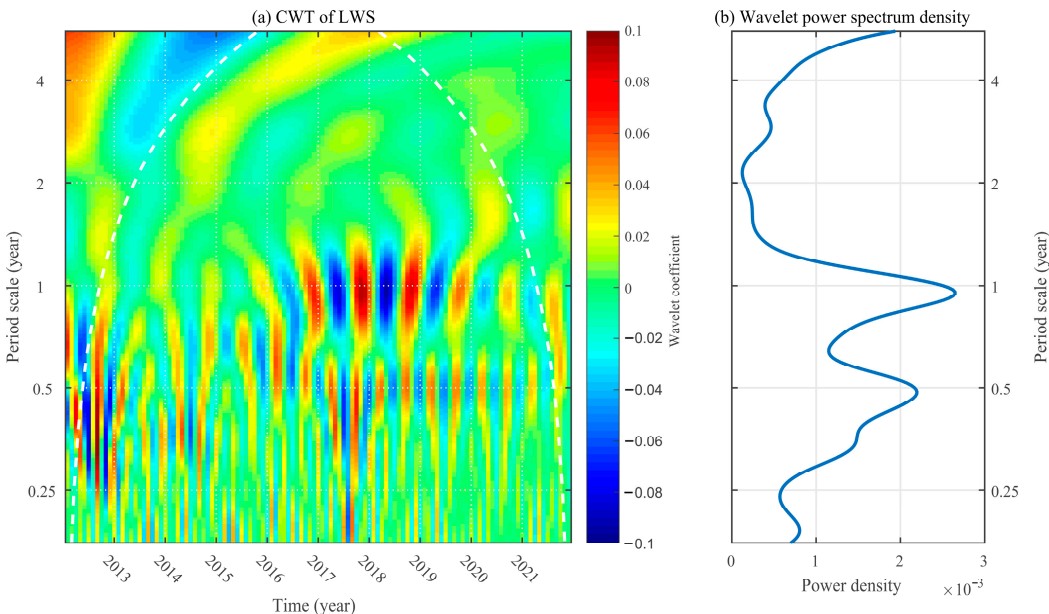

**Figure 9.** Wavelet analysis of the LWS time series of the KL from 2012 to 2021 using the Morlet wavelet: (**a**) continuous wavelet transform coefficient, (**b**) wavelet power spectrum.

The key advantage of wavelet analysis is that it can simultaneously be localized in the time and frequency domain. However, it cannot accurately determine the frequency of periodic components in a signal. In order to analyze the periodicity of the LWS time series further, we employed Fourier spectrum analysis. The power spectral density (PSD) estimated by Fourier analysis can be used to identify the accurate frequency of periodic variations in a time series [55,56]. Figure 10 shows the PSD of the LWS time series of KL, which is estimated using the MATLAB function 'periodogram'. The primary frequency of the LWS time series before the outburst (red line) is one cycle per year, while the highest peak appears at the frequency of two cycles per year in the spectral line after the outburst (blue line); this indicates that the outburst of ZL enhanced the seasonal variation of the LWS of KL. This finding is in accord with the results of wavelet analysis.

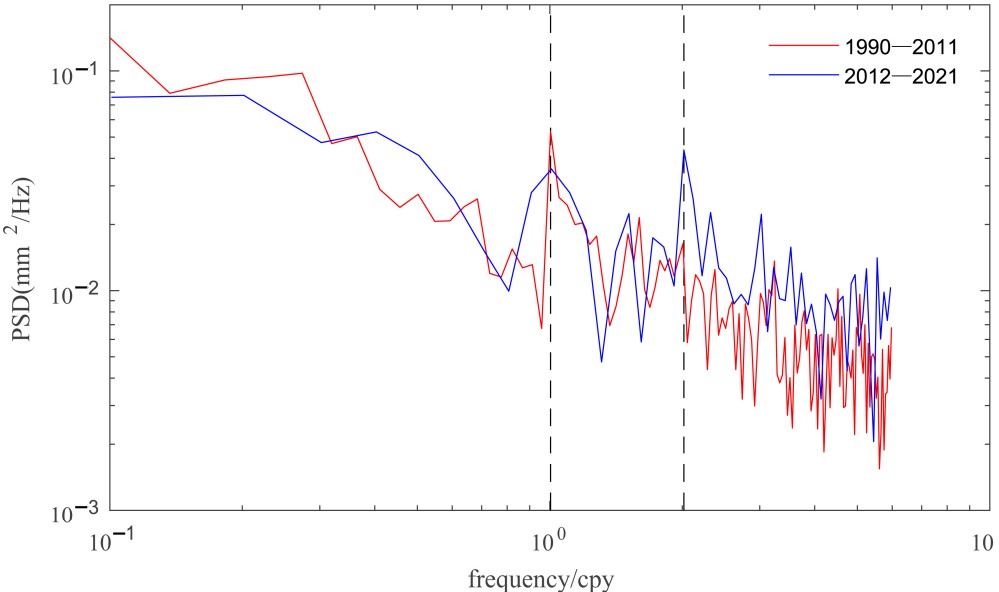

**Figure 10.** Power spectral density (PSD) of the LWS time series before (in red) and after (in blue) the outburst.

### 5.2. Annual LWS Variations before and after the Outburst

Figure 11 shows the months in which the maximum and minimum LWS of KL occurred from 1990 to 2021 and plots the maximum, minimum, and amplitude (the difference between the maximum and the minimum) of the LWS for each year. Figure 11a shows that the minimum LWS generally occurred in May or June before 2011, and the maximum LWS was observed between September and November. The abnormal occurrence of maximum in January 1994 and February 2004 can be attributed to the estimation error of LWS. It can be seen from Figure 7a that the annual variation of LWS before 2005 was small. Some months had very close LWS values, so their difference might be less than the estimation error. For instance, in 2004, the maximum LWS was 0.124 km³ in February, followed by 0.119 km³ in November. The difference is only 0.005 km³, which is smaller than the uncertainty of LWS.

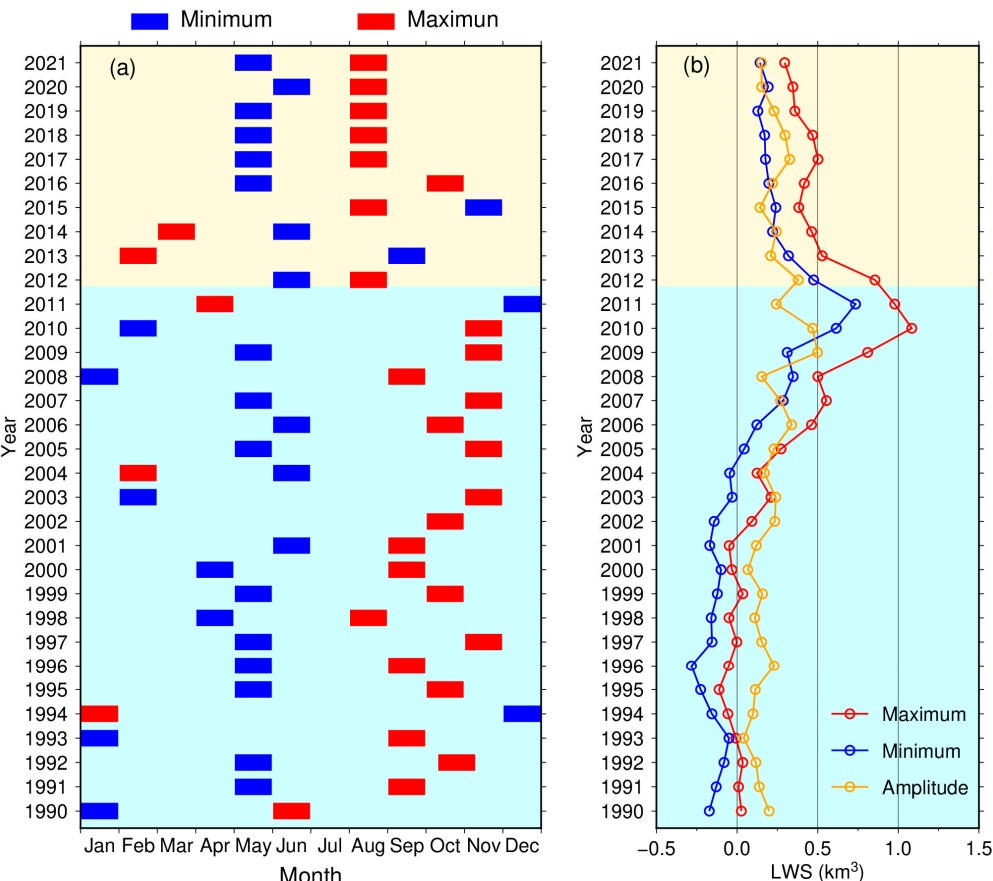

**Figure 11.** Annual LWS variations of KL from 1990 to 2021: (**a**) months in which peaks and troughs occur, and (**b**) the minimum, maximum, and amplitude of the LWS each year.

In the years (2012–2015) following the outburst, annual LWS variations became disordered. It can be explained by the alteration of the recharge and discharge pattern of KL. After the outburst, the input water from ZL's upstream and outflow to HN's downstream significantly affected KL's water budget. It disrupted the original law of LWS variation of KL when it was a closed lake. Until 2016, the occurrence of minimum LWS reverted to May. However, the occurrence of maximum LWS shifted forward to August.

Figure 11b demonstrates that the minimum and maximum LWS have an increasing trend from 1990 to 2021. The increasing trend reflects the correlation with climate change. In the past 30 years, annual precipitation in the ZL–SL basin increased with a growth rate of 2.29 mm/y [14]. Furthermore, it has been revealed that the total TWS in the basin was consistently increasing [34]. However, the outburst of ZL triggered the overflow of KL, reversing the increasing trend of TWS of KL, which resulted in a peak between 2009 and

2012. It can also be observed that the annual amplitude varies over a steady period of about 6 years. However, the mean magnitude of the annual amplitudes after the outburst increased to 0.244 km$^3$ from 0.168 km$^3$ before the outburst. It implies that the hydraulic connectivity among the lakes caused by the outburst significantly amplified the annual fluctuation of the LWS in KL.

*5.3. Possible Driving Factors of Pattern Change in the TWS Variation*

Figure 12 provides a comprehensive depiction of evaporation (E), precipitation (P), and the changes in glacier area (ΔG) and LWS (ΔLWS) in the KL basin from 1990 to 2017. The left panel showcases the monthly time series for these variables while showing the average monthly values before and after the outburst event. In terms of evaporation, it exhibits variability throughout the year. The evaporation mainly ranges between 60 mm and 180 mm, with higher values occurring during summer; this implies that evaporation rates are generally higher during the warmer months, which is consistent with the increased heat and solar radiation during this time. Precipitation is mainly concentrated from May to September, accounting for more than 92% of annual rainfall. Overall, these observations highlight the seasonal patterns and variations in rainfall and evaporation in the KL basin, with a concentration of precipitation in the summer months and higher evaporation rates during the warmer seasons.

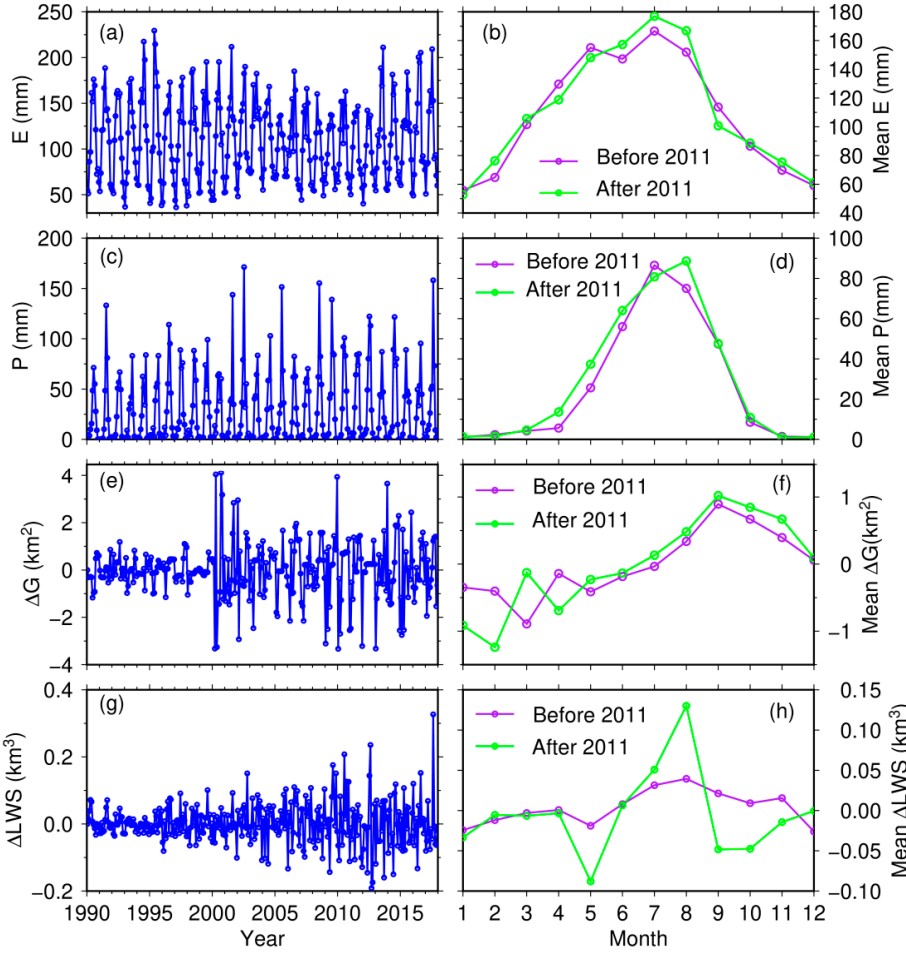

**Figure 12.** Monthly evapotranspiration (**a**,**b**), precipitation (**c**,**d**), change of glacier area (**e**,**f**), and LWS variation (**g**,**h**) in the KL basin. The left panel is the time series from 1990 to 2017. The right panel is the average monthly values before and after the outburst.

The glacier area within the basin, interpreted from remote sensing imagery, has a recurring pattern of expansion from June to December each year, followed by a melting

phase extending from January to June of the subsequent year. When considering the average monthly values, Figure 12b,d,f reveal that there are no significant changes in precipitation, evaporation, and glacier area before and after the outburst event.

However, Figure 12h demonstrates a remarkable difference in LWS. Based on the average monthly ΔLWS, it can be observed that there was a gradual increase in the LWS of KL from June to November before the outburst, followed by a gradual decrease from December to May in the following year. However, after the outburst, there was a quite different pattern in the LWS, with a rapid increase from May to August and a swift decline from September to October. The LWS variation after the outburst has a noticeable seasonal oscillation and a larger annual amplitude than before, corroborating the findings presented in Figure 11.

Based on the above analysis, it can be concluded that evaporation, rainfall, and glacier change are not the driving factors for the observed changes in TWS behavior after the outburst. The alteration of the local hydrological conditions caused by the outburst event may be the primary reason for these changes.

## 6. Conclusions

In this study, the water level time series was derived at KL using ERS-2 and Jason-1/2/3 altimeter data from May 1995 to April 2022. The lake area time series was determined by interpreting Landsat-5/7/8 images from January 1990 to November 2021. Combining these two series yielded an unprecedented long and high temporal resolution LWS time series based on the LA curve extracted from topographic data. The constructed LWS time series contains 1051 samples spanning from 1990 to 2022, indicating a high data density of 35.7 samples per year. The level, area, and LWS time series were validated by comparing three published datasets, demonstrating that this paper provides extended time series with enhanced temporal resolution and comparable accuracy.

Based on the high-temporal LWS time series, we characterized the water volume variations in KL before and after the ZL outburst in 2011 and highlighted the impact of the outburst. The outburst caused the water volume of KL to rapidly increase by ~2.76 km$^3$ in September 2011 and decrease by ~0.53 km$^3$ in October and changed the increasing trend of the lake level, which has been observed in many previous studies [15,34,57]. For the first time, this study revealed a significant change in the periodic patterns of LWS variations in KL. Prior to the outburst event, the dominant variations in the LWS time series were observed at 1-year and 3.5-year intervals, while no noticeable semi-annual signal was evident. However, after the outburst, the 3.5-year variation vanished, and a strong semi-annual oscillation became apparent. From 2012 to 2015, the outburst in ZL and the subsequent outflow from KL disrupted the periodic LWS variations in KL. Until 2016, regular cyclic signals have been restored with an amplified annual fluctuation. These findings suggest that the outburst event significantly impacted the periodic patterns of LWS variations in KL. It can be attributed to the alternation of hydrological conditions surrounding KL resulting from the outburst.

**Author Contributions:** Conceptualization, Z.H. and H.W.; Methodology, Z.H., H.W. and X.H.; Software, Z.H., X.W. and L.D.; Validation, X.W., Z.Z., X.H. and H.Z.; Formal analysis, X.W., Z.Z., L.D. and H.Z.; Investigation, Z.Z., L.D. and H.Z.; Resources, X.W.; Data curation, L.D.; Writing—original draft, Z.H., X.W. and Z.Z.; Writing—review & editing, H.W.; Visualization, Z.H. and X.W.; Supervision, H.W. and X.H.; Project administration, H.W.; Funding acquisition, H.W and Z.H. All authors have read and agreed to the published version of the manuscript.

**Funding:** This work was sponsored by the National Natural Science Foundation of China (41974016, 42104023, 42264001, 42364002), the Open Research Program of Key Laboratory of Marine Environmental Survey Technology and Application, Ministry of Natural Resources (MESTA-2020-A004), the Major Discipline Academic and Technical Leaders Training Program of Jiangxi Province (20225BCJ23014), Hebei Water Conservancy Research Plan (2022-28), the High-Level Talent Aggregation Project in Hunan Province, China-Innovation Team (2019RS1060).

**Data Availability Statement:** Publicly available datasets were used in this study. The datasets presented in this study can be found in their online repositories. The ERS-2 altimeter data were distributed by the European Space Agency (ftp://ra-ftp-ds.eo.esa.int (accessed on 28 August 2016)), and the Jason-1/2/3 SGDR altimeter products were produced and distributed by Aviso+ (ftp://ftp-access.aviso.altimetry.fr (accessed on 13 May 2022)). The Landsat-5 Thematic Mapper (TM), Landsat-7 Enhanced Thematic Mapper (ETM), Landsat 8 Operational Land Imager (OLI) data, and SRTM DEM can be found here: https://earthexplorer.usgs.gov (accessed on 31 August 2022); the ERA5 model data can be found from here: https://cds.climate.copernicus.eu/cdsapp#!/dataset/reanalysis-era5-single-levels?tab=overview (accessed on 10 May 2022); the DAHITI datasets can be found from here: https://dahiti.dgfi.tum.de/(accessed on 20 January 2023); The precipitation and evaporation data at Wudaoliao station can be downloaded from the China Meteorological Data Sharing Network at http://data.cma.cn (accessed on 31 December 2020).

**Acknowledgments:** We are grateful to ESA and AVISO for the altimeter data, USGS and DLR for remote sensing images, and SRTM data, ECMWF for providing ERA5 climate data. The meteorological data comes from China Meteorological Administration(CMA), and the pictures are drawn using GMT-5.2.1 and Matlab R2017a software.

**Conflicts of Interest:** The authors declare no conflicts of interest.

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
