# Peer review of "Characterizing the Water Storage Variation of Kusai Lake by Constructing Time Series from Multisource Remote Sensing Data"

_remotesensing, doi:10.3390/rs16010128_

Round 1
Reviewer 1 Report
Comments and Suggestions for Authors
The outburst event of large lakes is a rare phenomenon. In particular, the Zhuonai Lake outburst event is a sudden natural disaster under the background of warm and humidification on the Qinghai-Tibet Plateau, and is of great significance for studying the lake's response to climate change. they used satellite altimetry, remote sensing images and digital terrain models to conduct detailed data processing work to construct a high-resolution time series of Kusai Lake water storage, and observed lake mutations and seasonal signals after the lake broke its banks . The topic of this paper has a certain degree of novelty. It proposes an innovative data processing algorithm and analyzes the changes in water storage before and after the lake bank breaks based on high-resolution lake water storage time series, which has high research value.
1. There are certain differences in the meanings of LWS and ΔLWS. There are many suspected word mixing problems in the paper, such as lines 101, 177, 259, etc. Please check the full text carefully to avoid improper wording.
2. Figure 1 introduces the exact location of the Wudaoliang hydrological station that should be drawn in the study area.
3. This paper observes that there is an obvious semi-anniversary signal after 2012. This is a very interesting phenomenon, but it needs to be explained why this semi-anniversary signal appears.
4. ICESat on line 102 does not write the completed name
5. The time (month) expression in table1 is not standardized
6. ΔV in Formula 2 and Formula 3 can be changed to ΔLWS, which can unify the expression of the whole text.
7. In Section 4 (Results and Validation), the accuracy used to compare the data should be presented
8. Line 516 states that "data density is 36 per year", but in Table 3 it is 35.7 per year. It is recommended to modify the data in line 516.
9. The journal names of some documents in the references are too simplified, making it difficult to locate the accurate journal, such as documents 28, 51, etc.
Comments on the Quality of English LanguageEnglish language needs polishing.
Reviewer 2 Report
Comments and Suggestions for Authors
The manuscript titled as " Characterizing the Water Storage Variation of Kusai Lake by Constructing High-temporal-resolution Time Series from Multisource Remote Sensing Data: before and after the 2011 Outburst Event" can be used as a reference for future studies. However, the authors must address the following issues to make it suitable for publication. Thus, these concerns must be appropriately addressed and then reviewed again.
Ü The manuscript has numerous grammatical and methodological issues that must be checked and restructured to improve clarity, content, and organization.
Ü The manuscript's marginal scientific contribution/ nobility is unclear in the abstract or the introduction part. What is the marginal scientific contribution/ nobility of your manuscript?
1. Title
The title is too long that it needs revision. Thus, I suggest "Characterizing the Water Storage Variation of Kusai Lake by Constructing Time Series from Multisource Remote Sensing Data" because the monthly data presented in this study is not high-temporal resolution.
2. Figures
Figures (i.e., images and maps) must have a scale, a north arrow, and coordinates. The map in Figure 1 lacks the "North arrow," a primary cartographic parameter. Similarly, the red line used to represent Jason and the inset map of sub-basin is confusing. I suggest using different colors to avoid confusion for potential readers of your manuscript.
3. Methodology
· The satellite altimeter used in this study has different Along-track resolutions and vertical, Available, and Repeat cycles. How did you deal with these systematic differences when using these data together?
· In line 251 & Table 3, what do you mean by "Remote sensing satellite "and satellite imagery should be explicitly mentioned? Is it Landsat or other? Is it eight years? Make it straightforward for potential readers of your manuscript.
· In line 251, what do you mean by "Remote sensing satellite "should be explicitly mentioned? Is it Landsat or other? Make it clear for the potential readers of your manuscript.
· It is not clear how the Landsat-driven area was calculated. Please give a detailed methodology of how you derived it. If necessary, include also the area estimation's accuracy assessment in sight of the LULC analysis approach.
· In Line 417, you mentioned using "Fourier analysis". Where is the result of it? The manuscript lacks clarity in procedures or methodology.
4. Result
· In Figures 8 & 9, you have presented the result of wavelet analysis. How was the wavelet analysis done? I can't find an explanation regarding this analysis in your manuscript except for mentioning its name. I also suggest to explain why you did do this analysis.
· In Figure 10, you have presented the "power spectra of the LWS time series". How power of spectra of the LWS TS was calculated? In this research, one critical weakness is presenting results and graphs without explaining the methodology or introduction.
· In Figure 11:
o Why did the maximum occur in 1994, 2004, and 2011-2014 from January to April, while it occurred after August during other periods?
o The minimum, maximum, and altitude peaks that occurred between 2009 and 2012 should be explained, and if there is any link with global warming/ climate change, it should be further elaborated.
Minor edits;
1. The full names and abbreviations in parentheses (see lines 25─26) shall done per the manuscript writing standard Database for Hydrological Time Series of Inland Waters (DAHITI).
2. In Lines 258-261, some symbols are not visible. Make it clear.
.
Comments on the Quality of English Language
Moderate editing of English language required
Reviewer 3 Report
Comments and Suggestions for Authors
The manuscript focuses on investigating how the outburst event in 2011 affect the long-period variations of water volumes in Zhuonai Lake (ZL) using data from multisource remote sensors. The results are comprehensively analyzed. The manuscript is well organized and written.
Line 166: What is meant by “level” in “KL had a level of 4475 m”? Please explain it. What is the reference surface for the “level”? Is it the “lake level” better than “level”?
Line 211: What is the range of geoid gradients over the KL? This should be mentioned.
Line 211: “In this study, the dry and wet troposphere corrections were recalculated using the most recent climate models”. Which models? They need to be given.
Figure 6b: The reason behind the sudden jump of lake level from about 4480 m to ~4488m around 2012 should be further analyzed? Is there any sign/link of ENSO events?
